# Study on the Regeneration of Waste FCC Catalyst by Boron Modification

**DOI:** 10.3390/molecules29050962

**Published:** 2024-02-22

**Authors:** Chengyuan Yuan, Qiang Chen, Zhongfu Li, Jingyan Zhang, Conghua Liu

**Affiliations:** 1School of Materials Science and Engineering, Shandong University of Technology, Zibo 255000, China; lizhongfu@163.com (Z.L.); zhangjingyan@163.com (J.Z.); liuconghua@163.com (C.L.); 2Shanxi Tengmao Technology Co., Ltd., Hejin 043300, China; chengqiang@163.com

**Keywords:** waste FCC catalyst, regeneration, boron modification, acid sites accessibility, catalytic cracking

## Abstract

Regeneration has been considered as an ideal way for the post-treatment of waste FCC catalyst (ECat). In this work, the degeneration mechanism of ECat was firstly researched and attributed to the increasing of strong acid sites accessibility of ECat in contrast with fresh FCC catalyst by adsorption FTIR. Based on the proposed degeneration mechanism, ECat was successfully regenerated through suitable weakening for strong acid sites by boron modification. Characterization and evaluation results suggested that, the strong acid sites of regenerated ECat (R-ECat) were apparently decreased by boron modification which had significantly improve the heavy oil catalytic cracking performance of R-ECat. Because of the excellent performance, R-ECat in this work could successfully substitute for partial fresh FCC catalyst in FCC unit, which would provide a practicable way for the reutilization of ECat.

## 1. Introduction

Presently, fluid catalytic cracking (FCC) is still an important processing technology for lightening of heavy oil, in which FCC catalyst has played a key role and greatly influenced the performance of FCC unit [1,2]. As solid acid catalyst, FCC catalyst is usually consisted of zeolites, matrix and binder, in which zeolites are the critical active component and provide most of acid sites for FCC reaction [3]. In practical, FCC catalysts often severely degenerate after several runs in FCC unit because of the harsh condition of FCC reaction, which could seriously influence the running of FCC unit and thus bring about a great quantity of waste FCC catalyst (ECat) [4]. Therefore, the post-treatment for these waste FCC catalyst has become a serious issue for refinery and been paid more and more attention [5].

From the points of environment and economy, regeneration has been considered as an ideal post-treatment way for ECat, which could improve the catalytic cracking performance and thus realize the reusing of ECat in FCC unit [6]. Currently, the regeneration methods for ECat are mostly based on acid-extracting processes which could recover the structure properties and remove partial contaminated metals (V, Ni and Fe) for ECat [7]. For example, Cho et al. had removed contaminated metals from ECat by weak acid extracting and effectively enhanced the catalytic activity of ECat [8]. However, acid-extracting processes usually suffer from serious drawbacks such as complicated procedure, severe reaction condition and acid waste liquid discharge, which have extremely restricted their application in practice. Therefore, it is eagerly demanded to develop an effective regeneration method that with features of simple procedure, moderated reaction condition and environmental friendly for ECat. 

Based on the above fact, in this work, the degeneration for ECat was firstly discussed from the viewpoint of strong acid sites accessibility. According to the proposed degeneration mechanism, a novel and highly effective regeneration method through boron modification for ECat was developed. Furthermore, besides highly effective, the regeneration method in this work also possesses features such as simple procedure, low cost and environmental friendly in compared with conventional acid-extracting process, which would provide a viable method to realize the reusing of ECat. 

## 2. Results and Discussions 

### 2.1. Research on the Degeneration of ECat

Mostly, the degeneration of ECat has been attributed to the decreasing of acid sites that mainly caused by the destruction of zeolites in ECat in compared with fresh FCC catalyst [9]. Therefore, acid-extracting has been mostly used for the regeneration of ECat through enhancing the surface acidity of ECat [10]. However, strong surface acidity also could obviously promote the over-cracking and coking reactions of heavy oil, which would prominently increase the yields of dry gas and coke and thus seriously influence the catalytic cracking performance for regenerated ECat made by acid-extracting [11].

Different from the general understanding on the degeneration of ECat, the degeneration mechanism was firstly investigated from the point of strong acid sites accessibility of ECat in this work. Commonly, the micropore structure of zeolites in ECat is seriously destroyed in contrast with fresh FCC catalyst, which would obviously increase the strong acid sites accessibility of ECat for big heavy oil molecules. As a result, it is reasonable to infer that the coking and over-cracking reactions that easily-induced by strong acid sites over ECat would obviously increase in compared with fresh FCC catalyst and thus cause the degeneration of ECat.

To verify the above inference, adsorption FT-IR spectra were used to measure the strong acid sites accessibilities for fresh FCC catalyst and ECat with Py and DTBPy as probe molecules. As it reported, the areas of characteristic IR peaks for Py and DTBPy were proportional to the quantities of corresponding acid sites [12]. Here, the IR peak area for Py (A_Py_) after desorption at 400 °C is used to represent the quantity of total strong acid sites because of the small size of Py. Because of the big molecule size, the IR peak area for DTBPy (A_DTBPy_) after desorption at 400 °C is used to represent the quantity for the strong acid sites that could be accessed by big heavy oil molecules. Therefore, the value of A_DTBPy_/A_Py_ could be used to compare the strong acid sites accessibilities for different catalyst samples in the fact that bigger A_DTBPy_/A_Py_ would mean higher strong acid sites accessibility [13]. 

Figure 1 displays the Py-FTIR and DTBPy-FTIR spectra after desorption at 400 °C for fresh FCC catalyst and ECat. As it shown, the characteristic peaks at 1540 cm^−1^ and 3500 cm^−1^ could be attributed to the strong B acid sites that adsorbed by Py and DTBPy respectively [14]. 

The A_DTBPy_/A_Py_ values derived from Figure 1 for fresh FCC catalyst and ECat are shown in Figure 2. As it shown, the A_DTBPy_/A_Py_ value for ECat is much bigger than that of fresh FCC catalyst, indicating that the strong acid sites accessibility of ECat for big heavy oil molecules would be obviously higher than that of fresh FCC catalyst, which fully demonstrates the above inference on the degeneration of ECat. 

### 2.2. The Regeneration Mechanism of Boron Modification for ECat

Based on the above results, if the strong acid sites of ECat could be weaken suitably, the catalytic cracking performance of ECat would be effectively improved to achieve the regeneration for ECat duo to the decreasing for over-cracking and coking reactions in FCC process. 

According to previous reports, as it shown in Figure 3, boron could react with framework Al species of zeolites in calcination, which had decreased the acid sites quantity of (especially for strong acid sites) of zeolites [15,16]. Therefore, in this work, boron modification was adopted for the regeneration of ECat through decreasing the strong acid sites quantity of ECat. 

### 2.3. Characterization Results

The physiochemical properties for ECat and R-ECat are listed in Table 1. As it shown, the surface area, pore volume and contaminated metals contents (V, Ni and Fe) for R-ECat are not much different with that of ECat parent, suggesting that the regeneration of ECat by boron modification in this work are not based on the alteration of the above physiochemical properties. 

Figure 4 exhibits the NH_3_-TPD profiles of ECat and R-ECat. As it shown, both ECat and R-ECat display two main NH_3_-desorption peaks at around of 200 °C and 400 °C respectively, which could be attributed to moderated acid sites and strong acid sites respectively [17]. Compared with ECat, the NH_3_-desorption peak area at around of 400 °C for R-ECat obviously decreases, indicating the strong acid sites of R-ECat are successfully weaken by boron modification. 

The Py-FTIR spectra for ECat and R-ECat at different desorption temperature are shown in Figure 5. As it shown in Figure 5, the IR peaks at 1450 cm^−1^ and 1540 cm^−1^ for ECat and R-ECat could be attributed to L acid sites and B acid sites respectively. It can be seen from Figure 5a that, the IR peaks area of L acid sites and B acid sites after desorption at 200 °C for R-ECat are relatively closed to ECat, indicating that there is not too much difference in total acid sites quantity between ECat and R-ECat. Contrastively, as it shown in Figure 5b, the IR peaks area of L acid sites and B acid sites after desorption at 400 °C for R-ECat are visibly less than that of ECat, suggesting that the total quantity of strong acid sites for R-ECat is obviously decreased in compared with ECat because of boron modification. 

The specific acid sites quantities derived from Py-FTIR after desorption at 200 °C and 400 °C for ECat and R-ECat are listed in Table 2. It can be seen from Table 2 that, the quantities of L acid sites, B acid sites and total acid sites at 200 °C for R-ECat had slightly decreased in contrast with ECat. Compared with ECat, the strong acid sites quantities of L acid sites, B acid sites and total acid sites at 400 °C for R-ECat all obviously decreased, which is in good agreement with the result of NH_3_-TPD. 

### 2.4. Evaluation Results of Heavy Oil Catalytic Cracking

The heavy oil catalytic cracking performances for ECat, R-ECat and fresh catalyst are exhibited in Figure 6. As it exhibited, the dry gas, bottom and coke yields for R-ECat had decreased by 0.37%, 1.41% and 1.51% respectively in comparison with ECat, with gasoline, diesel and total liquid (LPG + gasoline + diesel) yields obvious increasing by 2.29%, 1.28% and 3.29% respectively. The above results fully demonstrate that the heavy oil catalytic cracking performance of ECat could be significantly improved by boron modification in this work, which would provide an effective way for the regeneration of ECat. 

The heavy oil catalytic cracking performances for ECat, R-ECat and fresh catalyst at different catalyst/oil mass ratios were also investigated. As it shown in Figure 7, the gasoline and total liquid yields of R-ECat are much higher than that of ECat at different catalyst/oil mass ratios meanwhile with obviously lower dry gas and coke yields, which further demonstrate the excellent heavy oil catalytic cracking performance of R-ECat. 

To confirm if the R-ECat could substitute for fresh FCC catalyst partially, the pure fresh FCC catalyst and blended catalyst that composited of 70% fresh FCC catalyst and 30% R-ECat were evaluated in one batch. As it shown in Figure 8, compared with fresh FCC catalyst, the heavy oil catalytic cracking performance of blended FCC catalyst is not visibly different, which indicate that the R-ECat in this work could be reused in FCC unit as substitute for partial fresh FCC catalyst without obvious influence on heavy oil catalytic cracking performance.

## 3. Materials and Methods

### 3.1. Materials

Fresh FCC catalyst and waste FCC catalyst (ECat) was provided by Shanxi Tengmao Technology Company. B_2_O_3_ was purchased from Sinopharm Chemical Reagent Company (Shanghai, China), analytically pure.

### 3.2. Regeneration of ECat

ECat was regenerated through boron modification using incipient wet impregnation method with B_2_O_3_ as boron source. After drying and calcination at 650 °C for 2 h, regenerated ECat was obtained (R-ECat). 

### 3.3. Characterizations and Evaluations

Pyridine (Py) and 2, 6-di-tert-butylpyridine (DTBPy) adsorption Fourier transform infrared spectra (FTIR) were recorded on a Burker TENSOR 27 instrument (Billerica, MA, USA). NH_3_ temperature programmed desorption. (NH_3_-TPD) was performed on a Micromeritics AUTOCHEM II 2920 chemisorption instrument (Micromeritics, Norcross, GA, USA) in the range of 100–500 °C at a heating rate of 15 °C/min. The adsorption of ammonia on the samples was performed at room temperature, followed by removing physically adsorbed ammonia at 100 °C for 1 h in flowing pure nitrogen. N_2_ adsorption–desorption measurement at −196 °C was performed on a Micromeritics ASAP 2010 instrument (Micromeritics instrument corporation, Norcross, GA, USA) to characterize the textural properties. The Brunauer–Emmett–Teller (BET) method and the Barrett–Joyner–Halenda (BJH) method were used to determine the surface areas and pore volumes of the samples. The boron content of R-ECat was determined by the inductively coupled plasma-atomic emission spectrometry (ICP-AES) method according to the procedures of US EPA 6010C (2007).

The heavy oil catalytic cracking evaluations for ECat and R-ECat on an advance cracking evaluation (ACE) unit developed by Kayser Technology Inc. Company, Houston, TX, USA, with the catalyst/oil weight ratio being 5. Gaseous products were analyzed using a GC-3000 online chromatograph produced by INFICON Company (New York, NY, USA) according to the UOP method 539. GC-3000 used four chromatographic modules for detection, and the work temperature was 0–50 °C. The carrier gases used were helium, hydrogen, nitrogen, and argon. Simulated distillation of liquid products was carried out using a 7890B chromatograph produced by Agilent Technologies Inc. (Santa Clara, CA, USA) according to the SH/T 0558 procedure. The working environmental temperature of Agilent 7890B was 15–35 °C. Retention time repeatability was <0.0008 min. Carrier and makeup gas settings were selectable for helium, hydrogen, nitrogen, and argon/methane. Coke deposited on the catalyst was quantified with a CO_2_ analyzer produced by Servomex Group Co., Ltd. (Sussex, England, UK). The detection range of the CO_2_ analyzer was 0–20%. When the detection value was lower than 0.4%, the regeneration step was considered to be completed. The conversion and yields of dry gas (H_2_ + C_1_ + C_2_), LPG (liquefied petroleum gas) (C_3_ + C_4_), gasoline (C_5_ < bp < 221 °C), diesel (221 °C < bp < 343 °C), bottom (bp > 343 °C), and coke were calculated. The properties of the heavy oil are listed in Table 3. 

## 4. Conclusions

(1)The degeneration mechanism of ECat was researched based on the strong acid sites accessibility by adsorption FTIR. The results suggested that the strong acid sites accessibility of ECat was obviously higher than that of fresh FCC catalyst, which would obviously promote the over-cracking and coking reactions in FCC process and thus cause the performance regeneration of ECat.(2)According to the above degeneration mechanism of ECat, boron modification was employed for the regeneration of ECat through weakening the strong acid sites of ECat because of the reaction between framework Al species in zeolites and B_2_O_3_.(3)Evaluation results demonstrated that, compared with ECat parent, the heavy oil catalytic cracking performance of regenerated ECat had been significantly improved, which made the regenerated ECat in this work could be as substitute for partial fresh FCC catalyst.(4)In comparison with current regeneration method for ECat, the regeneration method of boron modification in this work also possess advantages such as simple procedure, low cost and environmental friendly, which would make a good application prospect for it.

## Figures and Tables

**Figure 1 molecules-29-00962-f001:**
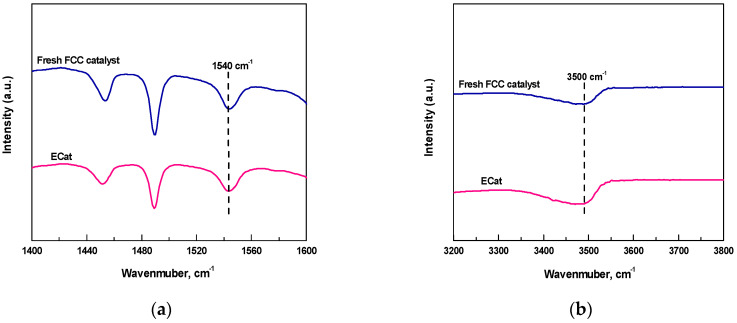
Py-FTIR (**a**) and DTBPy-FTIR (**b**) spectra for fresh FCC catalyst and ECat.

**Figure 2 molecules-29-00962-f002:**
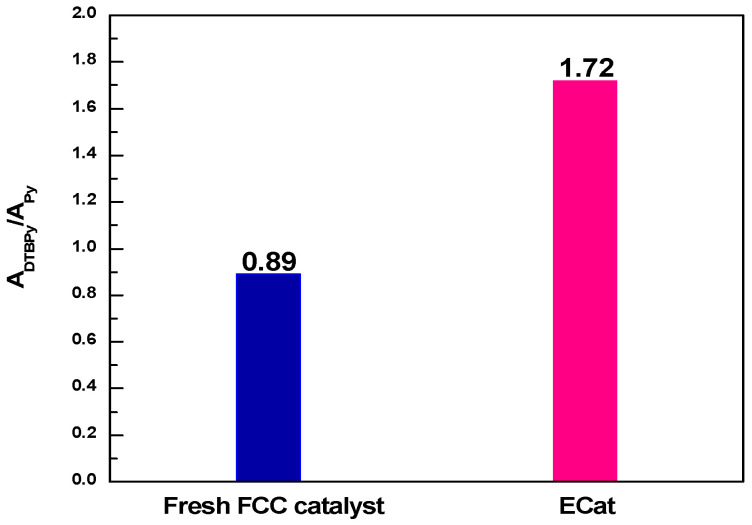
A_DTBPy_/A_Py_ values for fresh FCC catalyst and ECat.

**Figure 3 molecules-29-00962-f003:**
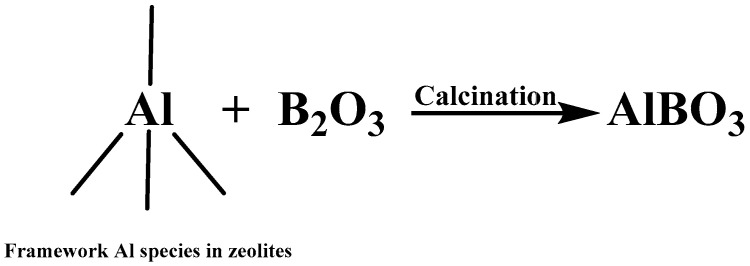
Reaction between framework Al in zeolites and boron.

**Figure 4 molecules-29-00962-f004:**
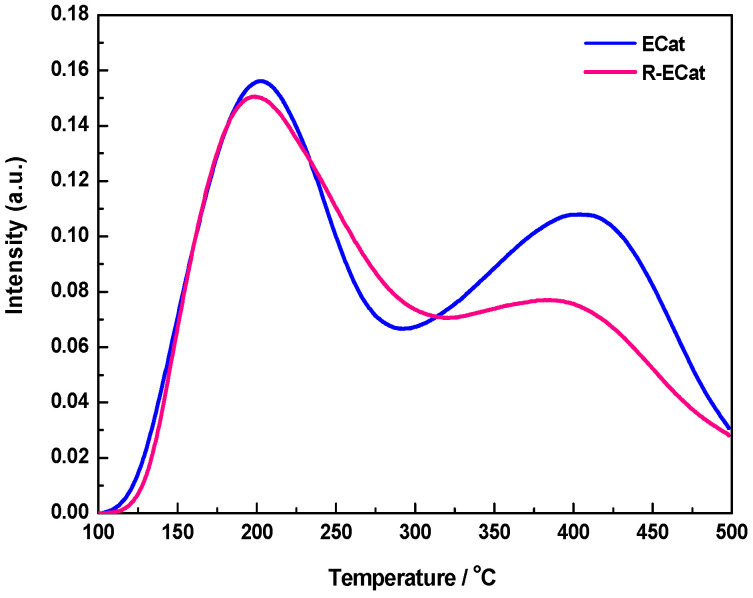
NH_3_-TPD profiles of ECat and R-ECat.

**Figure 5 molecules-29-00962-f005:**
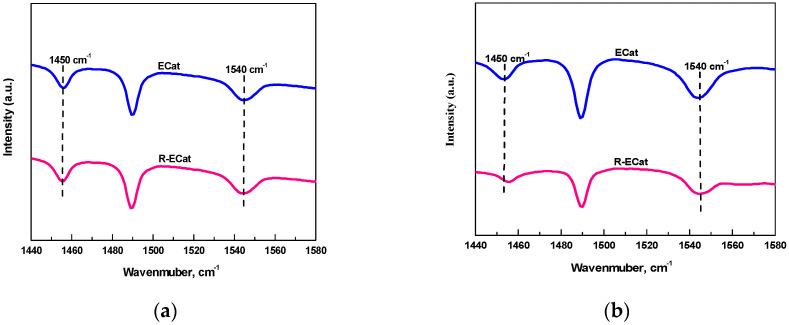
Py-FTIR spectra for ECat and R-ECat after desorption at 200 °C (**a**) and 400 °C (**b**).

**Figure 6 molecules-29-00962-f006:**
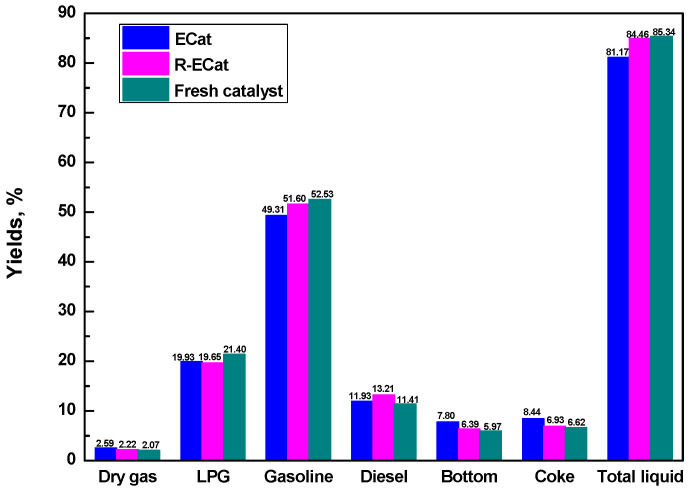
Heavy oil catalytic cracking performances for ECat, R-ECat and fresh catalyst.

**Figure 7 molecules-29-00962-f007:**
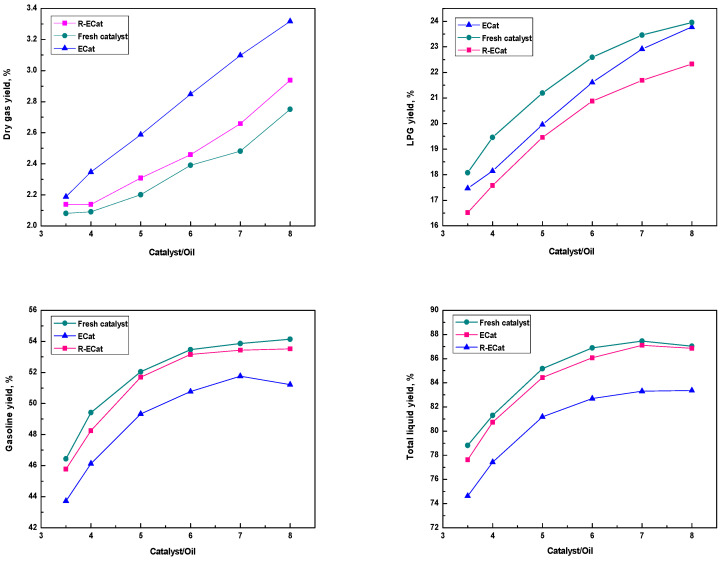
Heavy oil catalytic cracking performances for ECat, R-ECat and fresh catalyst at different catalyst/oil mass ratios.

**Figure 8 molecules-29-00962-f008:**
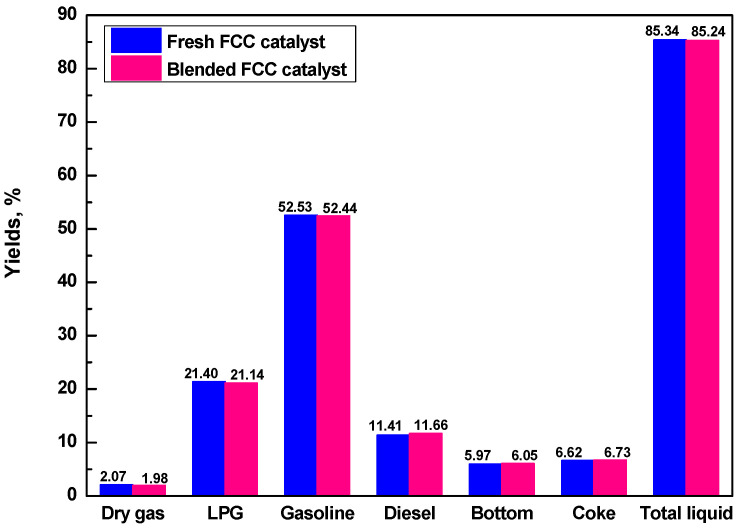
Heavy oil catalytic cracking performances for pure fresh FCC catalyst and blended FCC catalyst.

**Table 1 molecules-29-00962-t001:** Physiochemical properties for ECat and R-ECat.

Samples	Surface Area, m^2^/g	Pore Volume, cm^3^/g	V Content, μg/g	Ni Content, μg/g	Fe Content, μg/g	B Content, %
Fresh FCC catalyst	211	0.22	-	-	406	-
ECat	123	0.15	4045	7557	2028	-
R-ECat	120	0.14	4041	7550	2023	0.25

**Table 2 molecules-29-00962-t002:** Acid sites quantities of ECat and R-ECat.

Samples	Acid Sites Quantity (200 °C), μmol/g			Acid Sites Quantity (400 °C), μmol/g		
	L	B	Total	L	B	Total
Fresh FCC catalyst	74	147	221	17	31	48
ECat	50.7	112.8	163.5	9.4	19.2	28.6
R-ECat	44.2	105.6	149.8	4.3	11.5	15.8

**Table 3 molecules-29-00962-t003:** Properties of heavy oil.

Items	Values
Molecular weight/(g/mol)	374
Viscosity (100 °C)/(mm^2^/s)	12.27
Carbon residue/(%)	4.17
Metals/(μg/g)	
Fe	10.22
Ni	9.57
Ca	18.11
Cu	0.87
V	10.09
Pb	0.06
Na	16
Hydrocarbons/(%)	
Saturate	68.9
Aromatic	21.7
Resin	9.4

## Data Availability

Data are contained within the article.

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
