# Peer review of "Study on the Regeneration of Waste FCC Catalyst by Boron Modification"

_molecules, 2024, doi:10.3390/molecules29050962_

Round 1
Reviewer 1 Report
This work aims to regenerate waste FCC catalyst by boron modification, in order to overcome some drawbacks of current regeneration method, such as complicated procedure, severe reaction condition and acid waste liquid discharge.
The work is worth of publication after reviewing some of the following aspects and recommendations:
(1) I recommend to thoroughly revise English and some typos and vocabulary (for example, “percentage points” (maybe directly %).
(2) Figure 5. Py-FTIR spectra for ECat and R-ECat, the signal between 1400 and 1500 cm-1, to which species does it correspond or what could it indicate?
(3) The specific values of ADTBPy/APy should be added in Figure 2.
(4) What is “Total Liquid” in Figure 6. Please explain it on experimental or throughout the text.
(5) The procedure to calculate the coke content should be described in the manuscript.
(6) Please consider providing keywords that are not used in the article title.
The English can be further improved.
Author Response
This work aims to regenerate waste FCC catalyst by boron modification, in order to overcome some drawbacks of current regeneration method, such as complicated procedure, severe reaction condition and acid waste liquid discharge.
The work is worth of publication after reviewing some of the following aspects and recommendations:
(1) I recommend to thoroughly revise English and some typos and vocabulary (for example, “percentage points” (maybe directly %).
Reply: Thanks for your advice. We have made corresponding corrections in revised manuscript.
(2) Figure 5. Py-FTIR spectra for ECat and R-ECat, the signal between 1400 and 1500 cm-1, to which species does it correspond or what could it indicate?
Reply: Based on the report (Micropor Mesopor Mater, 2005, 82(1/2): 99-104) that, this signal could be attribute to the combined adsorption of B+L acid sites. However, the quantifies for B and L acid sites are usually made by 1540 and 1450 cm-1 respectively.
(3) The specific values of ADTBPy/APy should be added in Figure 2.
Reply: The corresponding values had been added in revised manuscript.
(4) What is “Total Liquid” in Figure 6. Please explain it on experimental or throughout the text.
Reply: The explanation of “Total Liquid” had been given in Figure 6 in revised manuscript.
(5) The procedure to calculate the coke content should be described in the manuscript.
Reply: The procedure to calculate the coke content had been added in in revised manuscript.
(6) Please consider providing keywords that are not used in the article title.
Reply: Thank you for your comments. We have made supplement in revised manuscript.
Reviewer 2 Report
The paper describes modification of ECat FCC catalyst by B2O3 to study the possibility of the reuse of wasted FCC catalyst. The paper can be useful for people working in the field of FCC catalysts and oil refining. There are some points needed to be improved.
First of all, it is not clear, what is the content of boron in the catalyst after impregnation. It should be clarified. Testing conditions, like temperature, has to be specified.
The second point is the comparison of catalytic activity of the catalysts. It is necessary to add the data for fresh catalyst to Figures 6 and 7.
There are some mistakes in English that have to be corrected. English must be checked thoroughly.
Author Response
The paper describes modification of ECat FCC catalyst by B2O3 to study the possibility of the reuse of wasted FCC catalyst. The paper can be useful for people working in the field of FCC catalysts and oil refining. There are some points needed to be improved.
First of all, it is not clear, what is the content of boron in the catalyst after impregnation. It should be clarified. Testing conditions, like temperature, has to be specified.
Reply: According to ICP result, the B content of catalyst was 0.25%。
The second point is the comparison of catalytic activity of the catalysts. It is necessary to add the data for fresh catalyst to Figures 6 and 7.
Reply: The performances for fresh catalyst had been supplemented to Figures 6 and 7 in revised manuscript.
There are some mistakes in English that have to be corrected. English must be checked thoroughly.
Reply: Thank you for your advice, the English of revised manuscript has been checked thoroughly.

Reviewer 3 Report
This work studied the the regeneration of waste FCC catalyst by boron modification. It would be considered for the publication in Molecules after a major revision.
1. It was necessary to distinguish the differences between fresh FCC catalyst, ECat, and R-ECat. Thus, the characterization and evaluation results of fresh FCC catalyst should be supplemented to Table 2-3 and Figure 4-7. It would help to confirm the role of boron modification.
2. A careful examination was strongly recommended to modify the grammar and expression.
A careful examination was strongly recommended to modify the grammar and expression.
Author Response
Reviewer 2
This work studied the the regeneration of waste FCC catalyst by boron modification. It would be considered for the publication in Molecules after a major revision.
- It was necessary to distinguish the differences between fresh FCC catalyst, ECat, and R-ECat. Thus, the characterization and evaluation results of fresh FCC catalyst should be supplemented to Table 2-3 and Figure 4-7. It would help to confirm the role of boron modification.
Reply: Thank you for your suggestion. The characterization results of fresh FCC catalyst had been supplemented in Table 2-3 of revised manuscript. The evaluation results of heavy oil catalytic cracking for fresh FCC catalyst had been displayed in Figure 8.
- A careful examination was strongly recommended to modify the grammar and expression.
Reply: Thank you for your suggestion. The grammar and expression had been carefully checked for revised manuscript.
Round 2
Reviewer 2 Report
I thank the Authors for the adding the testing data for fresh catalyst and checking English.
However, I am a little bit confused that they did not add boron content to the experimental part and did not add testing conditions as I asked.
Then, I recommend to accept the manuscript after the Authors make the corresponding corrections.
The quality of English is god enough.
Author Response
Thank you for your advice, the B content and its teasting method had been added into revised manuscript.
Reviewer 3 Report
The authors have answered the questions and modified the manuscript well, and thus it can be accepted.
Author Response
Thank you very much for your suggestion.